# Intermolecular epistasis shaped the function and evolution of an ancient transcription factor and its DNA binding sites

Dave W Anderson[1†], Alesia N McKeown[1†], Joseph W Thornton[2,3]*

[1]Institute of Ecology and Evolution, University of Oregon, Eugene, United States; [2]Department of Ecology and Evolution, University of Chicago, Chicago, United States; [3]Department of Human Genetics, University of Chicago, Chicago, United States

**Abstract** Complexes of specifically interacting molecules, such as transcription factor proteins (TFs) and the DNA response elements (REs) they recognize, control most biological processes, but little is known concerning the functional and evolutionary effects of epistatic interactions across molecular interfaces. We experimentally characterized all combinations of genotypes in the joint protein-DNA sequence space defined by an historical transition in TF-RE specificity that occurred some 500 million years ago in the DNA-binding domain of an ancient steroid hormone receptor. We found that rampant epistasis within and between the two molecules was essential to specific TF-RE recognition and to the evolution of a novel TF-RE complex with unique derived specificity. Permissive and restrictive epistatic mutations across the TF-RE interface opened and closed potential evolutionary paths accessible by the other, making the evolution of each molecule contingent on its partner's history and allowing a molecular complex with novel specificity to evolve.

*For correspondence: joet1@uchicago.edu

†These authors contributed equally to this work

Competing interests: The authors declare that no competing interests exist.

## Introduction

### Function and evolution in molecular sequence space

The relationship between gene sequence and molecular function is of central interest in both molecular biology and evolution. A useful construct for understanding this relationship is sequence space, an organized multidimensional representation of all possible genotypes of a biological system, each connected to its neighbors by edges representing changes in a single sequence site (*Maynard Smith, 1970*). Assigning functional information to each genotype yields a 'topological map' of the space, which depicts the total set of relations between sequence and function. As proteins evolve, they follow trajectories through sequence space, so the topology of the map also determines how mutation, drift, selection, and other forces can drive genetic evolution.

The functional topology of sequence space depends strongly on the degree and type of epistasis, defined as genetic interactions between sequence sites, such that the effect of a mutation at one site depends on the state at others. Epistasis makes the space's topology rugged (*Gavrilets, 2004*; *Kondrashov and Kondrashov, 2015*) in the sense that the functional effect of a mutation—a step in some direction along the map—depends on the genetic background in which it occurs. By causing the fitness effects of mutations to depend on the order in which they are introduced, epistasis can affect the probability that evolution will follow any given mutational trajectory under positive selection, purifying selection, or neutral drift (*Wright, 1932*; *Stadler et al., 2001*; *Weinreich et al., 2006*; *Poelwijk et al., 2007*; *Ortlund et al., 2007*; *Phillips, 2008*; *Bridgham et al., 2009*; *Field and Matz, 2010*;

**eLife digest** Transcription factors are proteins that control which genes inside a cell are active by binding to specific short sequences of DNA called response elements. Small differences in a trancription factor's amino acid sequence or in a response element's DNA sequence can affect their ability to recognize each other. How these pairs of molecules recognize each other—and how they evolved to do so—are important questions in molecular biology and evolution.

One way to understand these questions is to study 'sequence space', an organized representation of all of the possible sequences of a molecule, each linked to its neighbors by single point mutations. As a molecule evolves, it follows one of many possible paths from its ancestral form to a later-day version. Some paths deliver improvements at every step, and some involve 'neutral' wanderings. Still other paths produce intermediate forms that work poorly or not at all; such paths are unlikely to be followed during evolution. By characterizing many different versions of a molecule and mapping their functions onto sequence space, scientists can better understand how biological molecules work and how evolution might have produced them.

No one has previously explored the combined sequence space of two interacting molecules. Now Anderson, McKeown and Thornton have characterized the joint sequence space of a transcription factor that controls a cell's response to steroid hormones and the DNA response elements that it recognizes. Their experiments focused on the portion of sequence space between two ancient members of the transcription factor family which existed just before and just after a major shift in their ability to recognize different DNA sequences.

To reconstruct these ancestral proteins, Anderson, McKeown and Thornton used computational methods to infer their most likely sequences based on those of hundreds of present-day members of the family and the relationships between them. These proteins were then tested in the laboratory to see how strongly each of them could bind to various DNA response elements.

These proteins and their preferred response elements define the start and end of an ancient evolutionary journey through sequence space, which took place about 500 million years ago. Anderson, McKeown and Thornton then reconstructed all the possible steps on the paths between the two transcription factors and the two response elements. Every transcription factor was tested with every response element, and the information about how strongly they could bind was mapped onto the joint sequence space of the two molecules.

The experiments revealed that mutations in either the DNA or the transcription factor had very different effects, depending on which other changes had already occurred elsewhere in the same molecule or in its partner. Geneticists call this phenomenon 'epistasis'. Because of epistasis, only a handful of paths connected the ancestral protein-DNA complex to the derived complex without passing through intermediate steps that functioned poorly. These few likely paths all involved 'permissive mutations'—a change in the DNA that allowed the protein to tolerate a mutation that was previously detrimental, or vice versa.

The findings show that the evolution of each molecule depended critically on chance events in the evolutionary history of its partner. By changing the evolutionary potential of the molecule it interacted with, the members of the complex wandered through sequence space together. This journey yielded two new molecules that now work specifically together, each with functions that are distinct from their ancestors'.

*Salverda et al., 2011*; *Breen et al., 2012*; *Gong et al., 2013*; *Harms and Thornton, 2014*; *Yokoyama et al., 2014*; *Podgornaia and Laub, 2015*; *Tufts et al., 2015*).

Sequence space is vast, so an exhaustive functional mapping and characterization of epistasis for any protein or gene is impossible. Some studies have characterized libraries of genotypes in the sequence space immediately around present-day proteins (*Lunzer et al., 2005*; *Fowler et al., 2010*; *Hinkley et al., 2011*; *Araya et al., 2012*; *McLaughlin et al., 2012*; *Tokuriki et al., 2012*; *Jacquier et al., 2013*; *Melamed et al., 2013*; *Olson et al., 2014*; *Bank et al., 2015*; *Podgornaia and Laub, 2015*). Others have focused on smaller numbers of mutations that occurred during historical evolution, introducing them singly and in combination into extant proteins to understand their interactions and potential effects on evolutionary trajectories (*Weinreich et al., 2006*; *Elde et al., 2009*; *Bloom et al., 2010*;

*Gong et al., 2013*; *Natarajan et al., 2013*). Similar approaches have also been applied to reconstructed ancestral proteins in order to directly characterize how epistasis may have affected evolutionary history in the genetic backgrounds likely to have existed in the past (*Ortlund et al., 2007*; *Bridgham et al., 2009*; *Field and Matz, 2010*; *Harms and Thornton, 2010, 2014*; *Lynch et al., 2011*; *Yokoyama et al., 2014*; *Wilson et al., 2015*).

## Epistasis across a molecular interface: transcription factors and DNA response elements

Although the effect of epistasis on sequence space and evolution has begun to be characterized for individual molecules, many biological functions depend on physical interactions between molecules. Epistasis between sites across a molecular interface could play a key role in determining the functions and evolutionary potential of molecular complexes. An important but unexplored goal is therefore to functionally map the joint multidimensional sequence space that contains the combined genotypes of interacting molecules.

The interactions between transcription factors (TFs) and the DNA response elements (REs) to which they bind exemplify this issue. TF-RE interactions regulate gene expression in virtually all biological processes (*Tjian and Maniatis, 1994*; *Lelli et al., 2012*). Effective and precise gene regulation depends on the capacity of a TF to specifically bind its preferred RE targets with sufficient affinity and occupancy in a heterogeneous cellular environment (*Li et al., 2008*; *Fisher et al., 2012*). The genetic determinants of affinity and specificity of TF-RE complexes must lie in both molecules and the interactions between them: for example, an amino acid replacement that changes a TF's DNA specificity must affect affinity differently when combined with various RE genotypes (*Voordeckers et al., 2015*).

The joint sequence space of TFs and their REs has not been directly characterized, particularly in an evolutionary context. Previous work on TF-DNA recognition suggests that epistasis is likely to be important, as expected given the biophysical complexity of protein-DNA interfaces (*Dill, 1997*; *Stout et al., 1998*; *Pabo and Nekludova, 2000*). For example, numerous studies have assessed the binding of a single TF to a library of REs, thus identifying the genetic states in the DNA that determine affinity (*Badis et al., 2009*; *Portales-Casamar et al., 2010*; *Stormo and Zhao, 2010*; *Payne and Wagner, 2014*); in several cases, epistatic interactions between neighboring nucleotides in the RE are apparent (*Man and Stormo, 2001*; *O'Flanagan et al., 2005*; *Moyroud et al., 2011*; *Zhao et al., 2012*). Other studies have addressed aspects of the TF's protein sequence space by investigating how amino acid variation in a TF affects RE binding (*Lynch et al., 2008*; *Baker et al., 2011*; *McKeown et al., 2014*; *Perez et al., 2014*; *Pougach et al., 2014*), and here too there is some evidence of epistasis between residues (*Pabo and Nekludova, 2000*). We have little systematic knowledge, however, concerning the topology of joint TF-RE sequence space and how it may affect evolutionary processes.

The total joint sequence space of multiple molecules is far too large to characterize comprehensively. It should be possible, however, to functionally map the small portion of that space defined by a specific historical change in function. Such a region contains all genotypes on all direct mutational paths between the molecules in a reconstructed ancestral complex and those in a descendant complex that has different specificity. Here we map the joint sequence space across an evolutionary transition in specificity for an ancient TF protein and the DNA REs to which it binds. This approach allowed us to identify sequence states in the DNA and protein that determine the affinity and specificity of binding, to characterize epistasis within and between the molecules, and to analyze the effects of intermolecular epistasis on the evolution of gene regulation and TF-RE interactions.

## Evolutionary sequence space of an ancient steroid receptor and its REs

The DNA binding domain (DBD) of steroid hormone receptors (SRs) are a model for exploring the sequence space of an evolving TF-RE complex. SRs are a class of ligand-activated TFs; they include a ligand-binding domain—which activates gene expression in the presence of specific sex or adrenal steroid hormones—and a DNA-binding domain, which binds as a dimer to palindromic REs consisting of two half-sites each six bases long (*Bentley, 1998*; *Bain et al., 2007*). SRs group into two phylogenetic clades, each with a distinct DNA-binding specificity (*Figure 1A*). The estrogen receptors (ERs) preferentially bind to estrogen RE (ERE), which contain the half-site AG<u>G</u>TCA. The other clade—progestagen, androgen, mineralocorticoid and glucocorticoid receptors

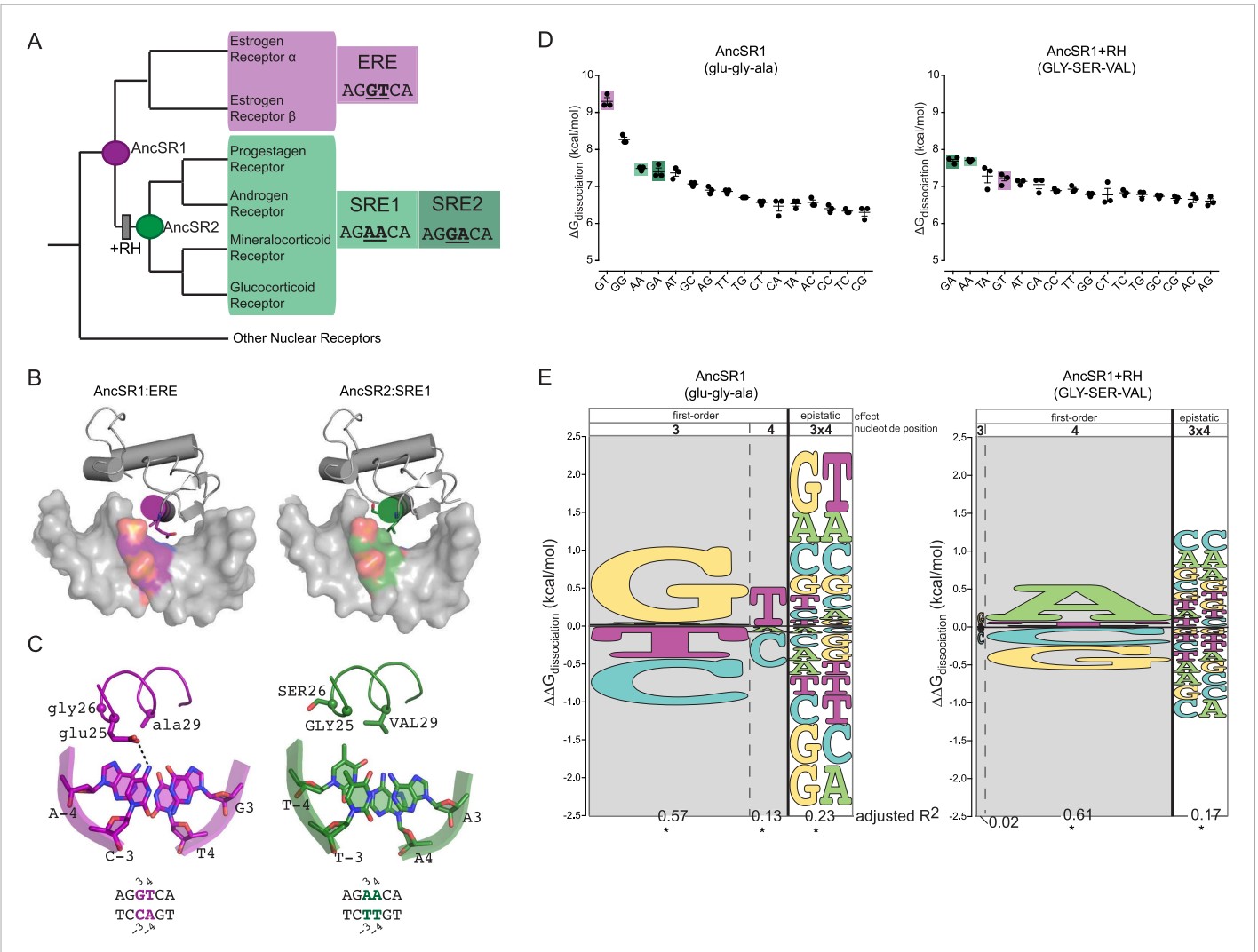

**Figure 1**. Recognition helix (RH) substitutions change DNA-binding affinity and specificity. (**A**) Phylogenetic relationships of modern-day vertebrate SRs are shown, with ancestral proteins AncSR1 and AncSR2 marked. Each protein's preferred response element (RE) is shown: estrogen RE (ERE; purple) or steroid REs (SRE1, SRE2; light and dark green, respectively), with the half-site sequence of each. Gray box indicates evolutionary interval in which SRE specificity evolved (*McKeown et al., 2014*). (**B**) Interface of steroid hormone receptor DNA-binding domains (DBDs) with their preferred RE half-sites. X-ray crystal structures of AncSR1 with ERE (left, 4OLN) and AncSR2 with SRE1 (right, 4OOR). The RH is shown as a colored cylinder; sticks, side chains that differ between AncSR1 and AncSR2. Colored surface, nucleotides that differ between REs. (**C**) Close-up of protein-DNA interface for AncSR1:ERE (left) and AncSR2:SRE1 (right). In the DBD, the RH is shown as ribbon, with side chains of variable amino acids shown as sticks and Cα as spheres. In the RE, variable nucleotides are shown as sticks with backbone as cartoon. Atoms are colored by element. Dashed lines, polar interactions between variable amino acids and nucleotides. (**D**) Historical RH replacements change AncSR1's affinity for REs. Binding energies of AncSR1 (left) and AncSR1+RH replacements were measured using fluorescence polarization to single half-site REs containing all possible combinations of nucleotides at the sites that vary between ERE and SREs. ERE, SRE1 and SRE2 are highlighted in purple, light green and dark green, respectively. $\Delta G_{dissociation}$ is the free energy of dissociation, calculated from dissociation constant ($K_d$). Technical replicates (dots) with mean and SEM (lines) are shown. (**E**) RH replacements change the genetic determinants of affinity within the RE. Energy logos for AncSR1 (left) and AncSR1+RH (right) show the effects of nucleotide states on binding energy relative to the average across all REs tested; states with $\Delta\Delta G_{dissociation} > 0$ are associated with higher affinity binding. Main effects of nucleotides at variable positions 3 and 4 are shown, as is the epistatic effect of nucleotide combinations, defined as the excess effect beyond that predicted under additivity. The height of each state indicates the magnitude of their effect on binding energy; states are ranked from top to bottom by the magnitude of its effect. Each column's width shows the portion of variation in binding energy attributable to the effects of states in that column, calculated as the increase in the adjusted $R^2$ when terms corresponding to those states are added to a linear regression model and fit to the experimental binding data. *, significant improvement in model fit (likelihood ratio test, p < 0.05 Bonferroni-corrected). For complete explanation of linear modeling approach, see 'Materials and methods'.

(PAMGRs)—preferentially bind to the SRE half-sites AGA<u>A</u>CA (SRE1) or AG<u>GA</u>CA (SRE2) (*Umesono and Evans, 1989*; *Lundback et al., 1993*; *Beato and Sanchez-Pacheco, 1996*; *Welboren et al., 2009*). Thus, the preferred RE half-sites for the two clades are identical except at two nucleotide positions (underlined).

The historical amino acid replacements that caused the two classes of extant SRs to evolve their distinct half-site specificities are already known (*McKeown et al., 2014*). We previously reconstructed the ancestor of all SRs (AncSR1) and the ancestor of all PAMGRs (AncSR2) and assayed their affinities for ERE and SREs (*Figure 1A*). We found that AncSR1 was ER-like, preferentially binding to ERE, and that AncSR2 was like PAMGRs, preferentially binding SREs. When just three of the 38 amino acid replacements that occurred during the evolutionary interval between AncSR1 and AncSR2 were introduced into AncSR1, they fully recapitulated the shift in half-site preference from ERE to SREs. These three replacements (glu25GLY, gly26SER, and ala29VAL, where lower and upper cases denote ancestral and derived states, respectively) were the only changes on the DBD's ten-residue recognition helix (RH), which inserts into the DNA major groove (*Figure 1B,C*). The ancestral and derived states at these three sites are completely conserved in modern-day ERs and PAMGRs, respectively, so the inference of their states in AncSR1 and AncSR2 was unambiguous. Although other substitutions during this interval affected the DBD's non-specific affinity for DNA and the cooperativity of dimeric binding, only the RH substitutions affected half-site specificity, and they did so without affecting cooperativity (*McKeown et al., 2014*).

The ancestral complex AncSR1:ERE and the derived complex AncSR1+RH:SRE define an ancient evolutionary transition in TF:RE specificity, and the set of direct paths between these points constitutes a historically relevant region of the joint protein-DNA sequence space. Here we report on experiments to functionally map this region of the molecules' joint sequence space and to understand the implications of its topology for evolutionary processes. We experimentally measured the binding affinity of AncSR1-DBD variants containing all genotypic combinations of ancestral and derived amino acids in the RH in complexes with REs containing all possible combinations of nucleotides at the two variable positions in the half-site. We used these data to statistically estimate the main effects of every variable state in the DBD and the RE and of interactions between states within and between each molecule. This analysis allowed us to investigate how the causal determinants of binding changed as the regulatory complex evolved during the historical shift in function and to evaluate the plausibility that any pathway through the joint TF-RE space might have been followed under various evolutionary scenarios.

## Results

### Exploitation of latent binding affinity

We first characterized the RE-specificity of the ancestral and derived proteins AncSR1 and AncSR1+ RH by measuring each protein's affinity for RE sequences containing all 16 combinations of possible nucleotides at the two sites that vary between ERE and SRE. We used a fluorescence anisotropy (FA) assay using labeled DNA probes, which provides direct and precise estimates of the free energy ($\Delta G$) of binding (*Figure 1D*, expressed as $\Delta G_{dissociation}$). Although the set of all possible REs is much larger, the preferred nucleotides at the other sequence positions in the RE did not change during SR evolution. We focused on differences in half-site affinity because the RH substitutions affected this phenomenon without changing the cooperativity of binding to palindromic REs (*McKeown et al., 2014*).

We found that several major changes occurred during the evolutionary transition from the ancestral to derived proteins. First, AncSR1 prefers ERE and AncSR1+RH prefers SREs, but both classes of RE are among the highest-affinity targets for both proteins (*Figure 1D*). The derived preference for SREs therefore arose by reshuffling the protein's relative affinities among high- and moderate-affinity targets, rather than by radically increasing affinity for DNA sequences that were previously bound very poorly.

Second, the RH substitutions dramatically impaired binding to the protein's best DNA targets in both absolute and relative terms; they reduced affinity for ERE by more than 2 kcal/mol while increasing affinity for SREs by a mere ~0.2 kcal/mol (*Figure 1D*). As a result, AncSR1's affinity for ERE is much higher than AncSR1+RH's affinity for SREs, and the difference between the best and next-best

targets in AncSR1 is much larger than that in AncSR1+RH. Together, these effects make AncSR1+RH less specific for its preferred REs than AncSR1.

The RH substitutions therefore changed the protein's preferred DNA binding sites by exploiting a preexisting latent affinity for a moderate-affinity RE and dramatically reducing affinity for the ancestral RE. Exploitation of latent, moderate- or low-affinity interactions has been observed in the functional evolution of other molecules (*Khersonsky et al., 2006*; *Coyle et al., 2013*; *Pougach et al., 2014*).

## Determinants of affinity in RE sequence space

We next used these measurements to quantitatively analyze the genetic factors within the RE's DNA sequence that determine affinity of AncSR1 and of AncSR1+RH. These determinants include the average (or 'main') effects of each possible nucleotide at the variable RE sites, as well as epistatic interactions, which occur when a combination of nucleotides at multiple sites affects affinity differently from expectation based on each nucleotide's average contribution. We used a multiple linear regression model (*Stormo, 2011*) that predicts a protein's free energy of binding to an RE as the sum of all main and epistatic effects at the two variable positions 3 and 4 (for details, see 'Materials and methods'); a linear model is appropriate because the dependent variable is the ΔG of reversible binding, which grows additively with the free energy of the factors that contribute to it (*Benson, 1968*; *Dill, 1997*; *Stout et al., 1998*). We used global regression to estimate the values of the model's parameters that best predict each protein's measured affinity for all 16 RE sequences. This approach allows us to quantify the contribution to the total binding energy made by each individual state (the main effects) and every combination of states (epistatic effects). It also allows us to estimate the portion of all variation in affinity explained by each main and epistatic effect (expressed as the increase in adjusted $R^2$ when a parameter for any class of effect is added to the model) and to evaluate the statistical significance of the improvement in fit attributable to each class of parameters.

We found that DNA affinity is determined by main effects of individual nucleotides and by epistatic interactions between them, but these determinants are radically different between AncSR1 and AncSR1+RH. AncSR1 prefers REs with G in position 3 (G3) by 1.0 kcal/mole and those with T in position 4 (T4) by 0.5 kcal/mole (*Figure 1E*, *Supplementary file 1*). A strong epistatic effect (G3xT4) is also present, which further enhances affinity for the GT combination by 0.8 kcal/mole beyond that expected from the main effects of these two nucleotides. This epistatic interaction establishes the protein's specificity for ERE by generating a large energy gap between binding to GT and binding to other sequences containing G3 or T4 but not both (*Figure 1D*). AncSR1's specificity is further affected by a strong negative epistatic determinant G3xA4, which substantially reduces its affinity for SRE2 (GA); without this interaction, the protein's preference for G3 would have made SRE2 (GA) a high affinity target (*Figure 1E*).

The DNA determinants of binding by AncSR1+RH are very different (*Figure 1E*, *Supplementary file 1*). Unlike its ancestor, this protein's affinity is unaffected by main effects of the nucleotide at DNA position 3. Further, the ancestral preference for T4 is replaced by a strong preference for A4. The ancestral epistatic determinant G3xT4 is nearly abolished and replaced by two new positive epistatic interactions—A3xA4 and G3xA4—which together establish the protein's preference for its best targets SRE1 and SRE2 relative to other sequences that contain A4. Consistent with the lower specificity of AncSR1+RH, the magnitude of all these determinants is smaller than those that determine AncSR1's target affinity.

These data indicate that in both the ancestral and derived proteins, epistasis between the two variable nucleotide positions in the DNA target played a key role in establishing specificity of binding. The evolutionary effect of the RH substitutions was to erase all major ancestral DNA determinants of affinity and to establish novel determinants, both main and epistatic.

## Evolutionary changes in TFs altered the DNA determinants of binding

We next sought to understand how each historical amino acid change in the RH altered the determinants of binding within the RE. We engineered variants of AncSR1 containing all combinations of ancestral and derived states at the three RH sites and measured each one's binding affinities for all 16 REs (*Figure 2*). All three RH amino acid replacements can be produced by single-nucleotide mutations; this set of proteins therefore comprises all direct pathways between AncSR1 and AncSR1+ RH and represents all of the most parsimonious possible evolutionary histories.

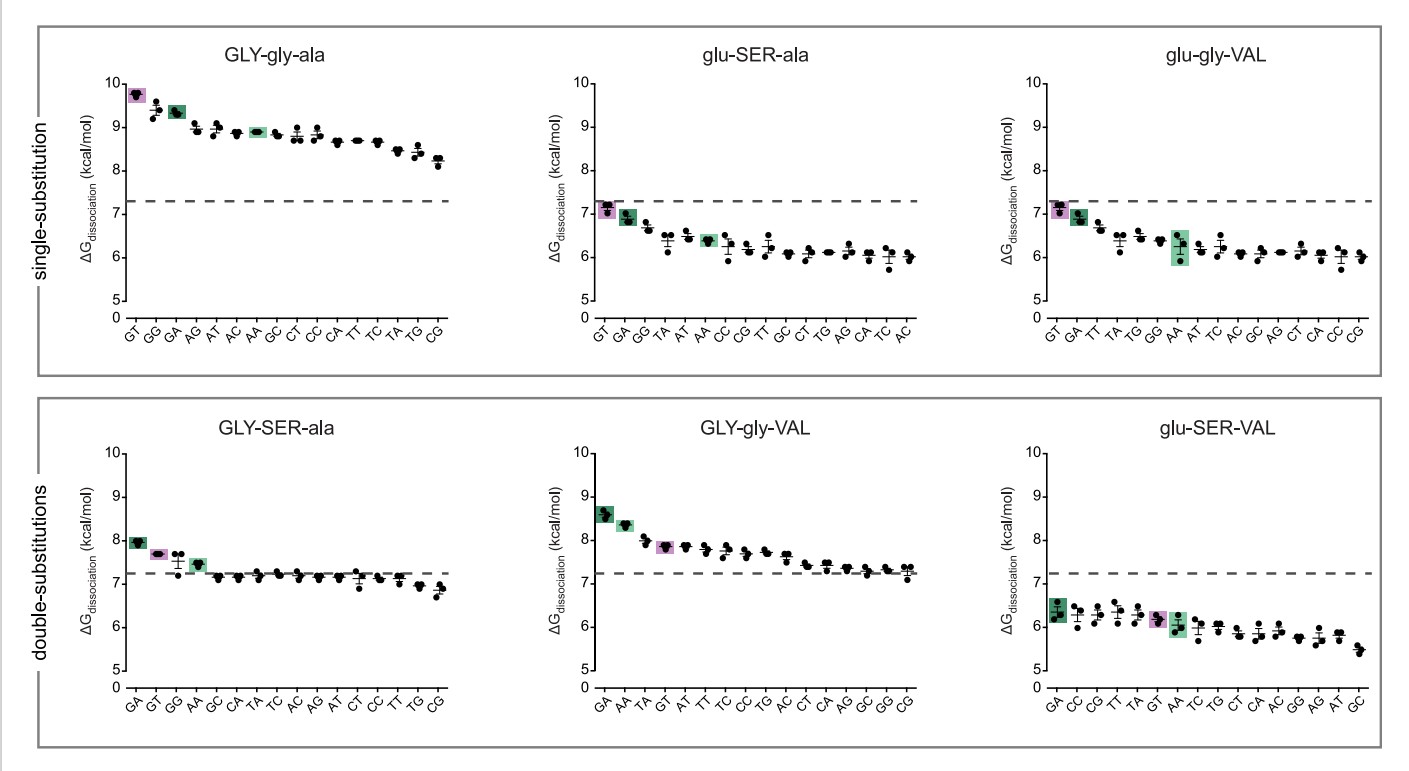

**Figure 2**. Protein intermediates between AncSR1 and AncSR1+RH are promiscuous or weak transcription factor proteins (TFs). Binding energies of AncSR1 variants containing all combinations of ancestral and derived states at the RH sites with historical replacements are shown for all 16 REs as measured by fluorescence polarization. Single-replacement neighbors of AncSR1 are shown in the top row; two-replacement proteins are shown in the bottom row. ERE, SRE1 and SRE2 are highlighted with purple, light green and dark green bars, respectively. Dashed line, mean binding energy across all protein genotypes and all REs. Data points show three replicates; mean and SEM are shown with lines.

Every intermediate protein we tested preferred either ERE or SREs over all other REs; transiently preferred targets do not emerge along trajectories from AncSR1 to AncSR1+RH. Unlike the starting and ending states, however, all intermediates were either universally low-affinity or highly promiscuous TFs. Specifically, three low-affinity intermediates (glu-gly-VAL, glu-SER-ala and glu-SER-VAL) bound to their best targets far more weakly than either the ancestral or derived forms bound their preferred sequences, and they did not bind any REs with affinity greater than the global average of all 8 proteins with all 16 REs (*Figure 2*). Two others (GLY-gly-ala and GLY-gly-VAL) promiscuously bound all 16 REs with greater-than-average affinity, and they both bound many targets—16 and 7, respectively—with affinity greater than AncSR1+RH's affinity for its best target. The remaining intermediate (GLY-SER-ala), was moderately promiscuous, binding four REs—ERE, SREs, and one other—with similar and above-average affinities.

We used these data to quantify the DNA determinants of TF affinity for each intermediate protein using the linear model described above. This allowed us to reveal how DNA specificity would have changed along any direct trajectory from AncSR1 to AncSR1+RH. We found that no single replacement is sufficient to abolish the ancestral DNA preferences or to establish the derived preferences. All of AncSR1's single-replacement neighbors maintain the major ancestral determinants of specificity—G3, T4, and G3xT4—but at reduced magnitude (*Figure 3*, *Supplementary file 1*), consistent with the fact that all three of these proteins prefer ERE, but to a lesser degree than AncSR1 does (*Figures 1D, 2*). None of the single-replacement neighbors display any of the derived determinants of specificity (main effect preference for A4 or epistatic preference for G3xA4 and A3xA4).

After the second step of the mutational pathway from AncSR1 to AncSR1+RH, some of the derived determinants of specificity begin to appear, and the ancestral determinants are further weakened

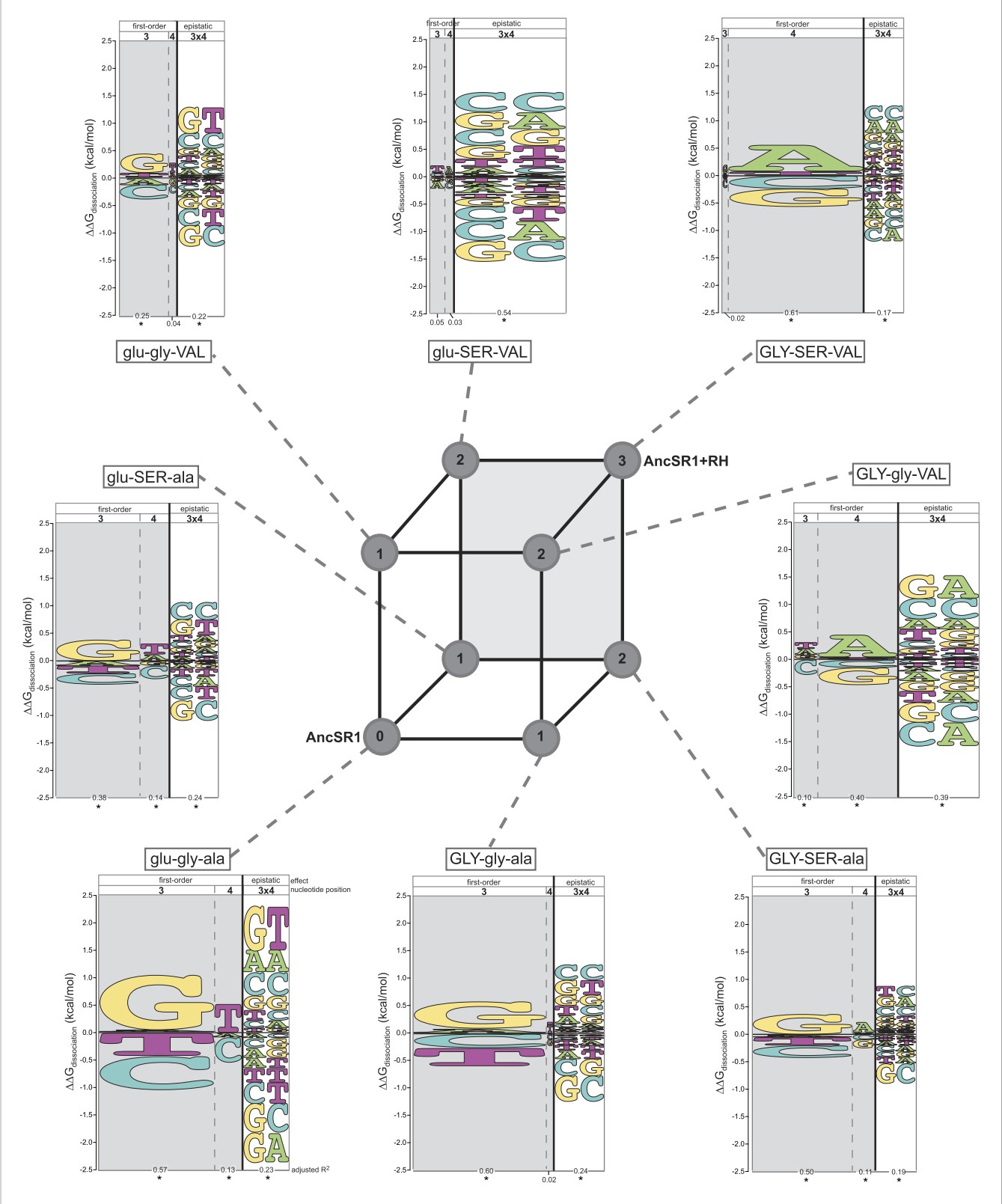

**Figure 3**. Each amino acid replacement contributes to the evolution of novel DNA specificity. For each protein intermediate in the sequence space between AncSR1 and AncSR1+RH, the energy logo depicts the main and epistatic effects of the RE nucleotide states and combinations on binding affinity by each TF (for details, see **Figure 1E**). Vertices of the cube indicate protein genotypes; the number of amino acid differences

*Figure 3. continued on next page*

*Figure 3. Continued*
from AncSR1 is indicated in the circle at each node. Edges represent single replacements between TF genotypes.

(*Figure 3*, *Supplementary file 1*). In all cases, however, either the ancestral determinants are partially retained or the derived determinants are weak. Only upon addition of the third replacement are the derived determinants complete. All three amino acid replacements in the RH therefore contributed to the derived DNA specificity, and their effects depend on the background into which they are introduced.

These analyses also point to a third-order form of epistasis: combinations of amino acid replacements uniquely affect binding to specific DNA sequences. The most apparent example of this phenomenon is the interaction between glu25GLY and ala29VAL in producing the derived preferences (*Figure 3*, *Supplementary file 1*). Neither replacement increases relative affinity for REs containing A4. But when the two changes are combined (GLY-gly-VAL or GLY-SER-VA)L, a strong A4 preference is established. Thus, the threefold combination of GLY25, VAL29, and nucleotide A4 increases affinity beyond that expected due to the main effects of any of these states or the pairwise interactions between them (see also *Supplementary file 1*).

Taken together, our findings indicate that all direct paths from AncSR1 to AncSR1+RH involve one of two scenarios: losing high-affinity binding to the ancestral and all other REs followed by a gain of binding to new targets, or gaining very promiscuous high-affinity binding followed by a dramatic narrowing of targets. No paths involve an immediate transformation of the highly specific ancestral TF into a specific protein with a new but narrow set of targets. Instead, ancestral determinants of binding were weakened and derived determinants gradually strengthened as mutational combinations were assembled.

## Effects of amino acid substitutions on affinity

We next quantitatively characterized the effects of each amino acid replacement on DNA affinity and specificity. For this purpose, we first expanded the regression model described above to incorporate variation in the TF's protein sequence and to determine the main and epistatic effects of each historical amino acid replacement on DNA affinity (*Supplementary file 2*).

We first analyzed nonspecific effects on affinity averaged across all REs tested. We found that each amino acid replacement changed non-specific affinity, but they did so in different directions. glu25GLY increased average affinity by 1.3 kcal/mol (*Figure 4A*, *Supplementary file 1*), consistent with the observation that GLY-gly-ala is a promiscuous and high-affinity DBD (*Figure 2*, *Supplementary file 1*). The other substitutions gly26SER and ala29VAL each reduced the average affinity of binding (*Figure 4A*, *Supplementary file 1*), explaining why proteins containing these states without glu25GLY are low-affinity proteins for all REs (*Figure 2*). Moderate epistatic interactions between the residues further modified their nonspecific effects on average affinity (*Figure 4A*, *Supplementary file 1*). When combined, the countervailing effects of the three replacements cause the final AncSR1+RH genotype to have an average affinity similar to that of AncSR1.

## Epistasis across a molecular interface

Differences among proteins in their average affinity over REs cannot explain the shift in DNA specificity that occurred between AncSR1 to AncSR1+RH. To change specificity, an amino acid replacement must affect affinity for RE genotypes differently. Therefore, when amino acid replacements change specificity or preference for REs, epistasis across the protein-DNA interface must be involved. To statistically characterize this form of epistasis, we further expanded the linear models described above to simultaneously analyze the effects of the combined protein + DNA genotype on affinity. This approach allowed us to identify epistatic effects on affinity caused by combinations of amino acid replacements and nucleotide states. The magnitude of an epistatic effect across the molecular interface is the difference in the binding energy of complexes containing both a specific TF amino acid and a specific RE nucleotide from that predicted based on the average effects of each of those states. A replacement may also participate in third-order interactions with

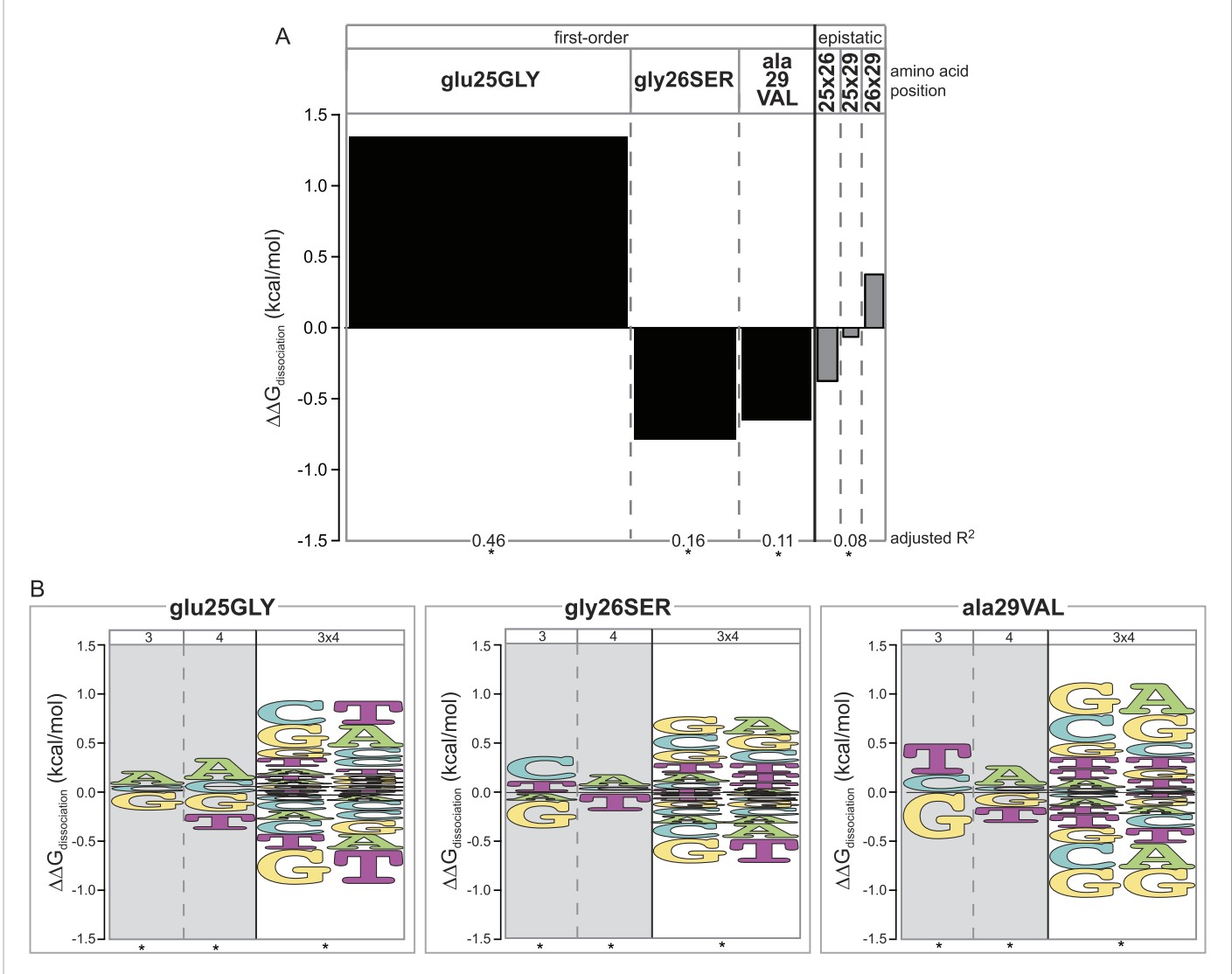

**Figure 4**. Epistasis across the protein-DNA interface: effect of historical replacements in the TF on DNA determinants of affinity in the RE. (**A**) Main and epistatic effects of RH replacement on DNA affinity. Bars indicate the mean change in binding energy caused by each amino acid change in the RH, averaged across all TF:RE combinations measured; epistatic effects represent the additional effect of pairs of replacements on average binding energy beyond that predicted by their main effects. Bar width depicts the portion of variation in binding energy attributable to each main or epistatic effect, calculated as the increase in the adjusted $R^2$ of the fit to the experimental binding data when each term is added to a linear regression model. *, significant improvement in model fit (likelihood ratio test $p < 0.05$ after Bonferroni correction). (**B**) Intermolecular epistasis. Energy logos indicate the effect of each amino acid replacement on the genetic determinants of binding within the RE. For each amino acid replacement, the size of each letter indicates the change the replacement causes in the main (or epistatic) effects of nucleotide states (or combinations) on relative binding energy.

a combination of nucleotide states in the RE, if it specifically changes relative affinity for the combination more than expected from the first- and second-order effects together.

We found that each amino acid replacement is involved in cross-interface epistatic interactions that contribute to eliminating the ancestral determinants of specificity and establishing the derived determinants. Although replacement glu25GLY increases affinity for all REs (*Figure 4A*), it does not do so uniformly: it increases relative affinity for REs with nucleotide states A3, A4 and G3xA4—thus specifically improving relative binding to SREs—and decreases relative affinity for REs containing the ancestral DNA determinants G3 and T4, as well as the ERE-specific combination G3xT4 (*Figure 4B*, *Supplementary file 1*). This replacement's most dramatic effect is therefore to cause a specific increase in the protein's affinity for SREs relative to EREs.

The other two replacements also participate in cross-interface pairwise and third-order epistatic interactions. Both gly26SER and ala29VAL reduce the ancestral determinants of binding G3 and T4 and increase the derived preference for REs with A4. Both also are involved in third-order interactions, by which the amino acid changes further reduce affinity for the G3xT4 combination and increase affinity for the G3xA4 combination (*Figure 4B*, *Supplementary file 1*). As a result, although each of these replacements reduces binding affinity to every single RE (*Figure 4A*), they do so most radically on ERE while only weakly affecting SREs (*Figure 4B*).

Taken together, these observations confirm that all three RH replacements contributed to the historical change in specificity via cross-interface epistatic interactions. Second-order intermolecular interactions allowed each replacement in the TF to shift specificity by differentially affecting affinity for REs containing specific individual nucleotide states. Third-order interactions allowed each replacement to have further effects on affinity when combined with pairs of nucleotides, beyond those expected based on any of the three sites' average effects plus all their pairwise interactions.

We found that new specificity evolved without sign epistasis. Each replacement affected binding to all 16 REs in the same direction (*Figures 2, 4*), but its effects on some REs were more extreme than on others. Because none of the replacements acted as a switch that impaired binding to some sequences while improving affinity for others, multiple replacements were necessary to achieve the derived specificity and affinity. Why would the shift in function not have occurred via a simpler genetic mechanism involving sign epistasis? We speculate that there are few or no potential replacements that have pinpoint opposite effects on different REs. The interface between a TF and DNA is heterogeneous and densely packed, and all four possible nucleotides share many similarities in the physical properties they confer on the DNA surface to which a TF binds. Thus, amino acid changes are more likely to alter binding in a generic direction across all possible versions of the RE, but they do so more effectively when paired with some nucleotides than with others.

## Mutational pathways to new transcriptional modules

Having mapped the free energy of binding—a fundamental biochemical property—across joint sequence space, we next sought to understand how evolution might proceed through this space under various evolutionary scenarios. If the relationship between biochemical affinity of a TF-RE complex and its function in the cell and organism were linear, then the ruggedness of the biochemical topology of sequence space would be identical to the ruggedness of the space's functional topology. It is highly unlikely, however, that all biological dependent variables—occupancy on DNA, gene regulatory output, phenotypic effect, and fitness—are linearly related the affinity of a TF-RE complex. A nonlinear relationship between affinity and biological function/fitness would introduce additional epistasis into the topology of sequence space and further change the kinds of paths that evolution is likely to follow under purifying selection, drift, and positive selection. The precise nature of this transformation depends strongly on biological context, is unknown. We therefore sought to gain preliminary insight into the plausibility of evolutionary trajectories through joint sequence space under two simple, biologically motivated scenarios.

The first scenario was to consider how AncSR1 might have evolved novel recognition of SREs, thus establishing occupancy of a new set of target genes, while under purifying selection to maintain specific, high-affinity binding to at least one RE at every step in the trajectory. We based our analysis on the concept of a connected network of functional protein genotypes, which assumes that purifying selection makes mutational pathways through nonfunctional intermediates very unlikely (*Maynard Smith, 1970*; *Wagner, 2008*). Although the RH substitutions occurred after a gene duplication of AncSR1, it is unlikely that either copy was released from purifying selection, because the ancestral function was conserved in both copies for tens or hundreds of millions of years before neofunctionalization occurred (*Bridgham et al., 2008*; *Eick et al., 2012*). We defined each protein's set of functional RE targets as those that fulfill two simple criteria. First, to achieve reasonable occupancy by the low concentrations of TF typical in cells (*Fisher et al., 2012*), the RE must be bound with moderate to high affinity, which we defined as greater than the average affinity across all TF-RE complexes tested. Further, a potential DNA binding site must compete with other REs for occupancy by the same TF, so the second criterion we imposed is that the affinity constant for an RE must be within a factor of ten of that protein's best target (*Fisher et al., 2012*). TF:RE complexes not meeting these criteria were classified as low-occupancy and therefore nonfunctional. Although these criteria

are somewhat arbitrary, they provide a starting point for understanding how the biochemical epistasis we observed, along with a simple nonlinearity in the transformation of affinity into function, might affect the mutational paths available to the evolving TF-RE complex.

We found that epistasis strongly structures possible evolutionary trajectories through sequence space. Starting from AncSR1, only one replacement (glu25GLY) leads to a functional TF; the others yield proteins that do not effectively bind any REs and are therefore unlikely evolutionary intermediates (*Figure 5*). The glu25GLY mutation produces a high-affinity but extremely promiscuous TF that retains strong binding to the ancestral ERE while gaining high-affinity binding to 13 novel REs, including the SREs. This replacement opens up new evolutionary trajectories, because once it is in place, either of the other two amino acid changes become tolerable. glu25GLY therefore represents a permissive evolutionary mutation that broadens the network of other accessible replacements that the protein can explore.

Epistasis also affects the second and third steps in the protein's mutational trajectory. The two possible paths available to AncSR1+RH have different functional implications for specific RE recognition, depending on the order in which the remaining replacements are introduced. Adding ala29VAL first expands the set of occupied DNA targets to include all 16 REs; the final gly26SER mutation then radically narrows the TF's specificity, eliminating 13 REs as high-occupancy targets and leaving only SRE1, SRE2, and one additional target (TA). Following the other pathway, incorporating gly26SER first narrows promiscuity somewhat but does not eliminate the ancestral targets, leaving four high-affinity REs (ERE and the other ancestrally recognized target GG, plus the novel SRE1 and SRE2); the final replacement ala29VAL then eliminates the ancestral targets—rather than expanding the protein's promiscuity, as it would if introduced earlier—and yields a protein specific for the derived REs (*Figure 5*). These observations point to higher-order epistasis—two amino acids interacting with each other to differentially change regulation of specific DNA sequences—which causes the functional implications of trajectories through sequence space to depend on the order of sequence change among multiple molecules.

Taken together, these data indicate that a derived TF could evolve to regulate a novel set of DNA targets completely distinct from those of the ancestral protein without losing the ancestral function until the final amino acid change. Because of epistasis—particularly the requirement for the permissive replacement glu25GLY for the other two replacements to be tolerated—only a small fraction of the possible pathways through protein sequence space pass through only functional TF intermediates on the way to the derived protein (*Figure 5*). Along both pathways, promiscuous intermediate genotypes gained recognition of the novel SREs, followed by further replacements that eliminated high-affinity binding to the ancestral RE and any transiently acquired targets. Our results are consistent with previous studies indicating an important role for permissive mutations in enabling a protein to tolerate other function-switching mutations that would otherwise be deleterious, particularly when a threshold relationship pertains between a biochemical property and fitness (*Ortlund et al., 2007*; *Bloom and Arnold, 2009*; *Woods et al., 2011*; *Gong et al., 2013*; *Harms and Thornton, 2013*). By strongly increasing the protein's affinity for many REs, replacement glu25GLY moved the evolving AncSR-DBD well above the threshold for functionality, allowing the protein to tolerate other replacements that refined the protein's specificity while decreasing its generic RE affinity.

## Intermolecular epistasis allows coevolutionary drift by the TF-RE complex

The second scenario we examined was a process of mutually permissive neutral drift by a functional unit of one TF and one RE under purifying selection. Numerous studies have found that a TF and its RE at a regulatory element sometimes diverge in sequence from their ancestral states while maintaining a conserved regulatory association with each other (*True and Haag, 2001*; *Haag and True, 2007*; *Barriere et al., 2012*; *Lynch and Hagner, 2015*). Using the same criteria as above to define a functional complex, we sought to understand whether a single complex of AncSR1:ERE could traverse the joint TF-RE sequence space through a series of changes in the protein and the DNA, reaching AncSR1+RH: SRE along a continuous neutral network without ever losing high-affinity, high-occupancy binding.

We found that there are many pathways through the joint genotype space from the ancestral to the derived complex that maintain a functional complex at every intermediate step. These pathways are made possible by intermolecular epistasis, which causes replacements in the TF to alter constraints on

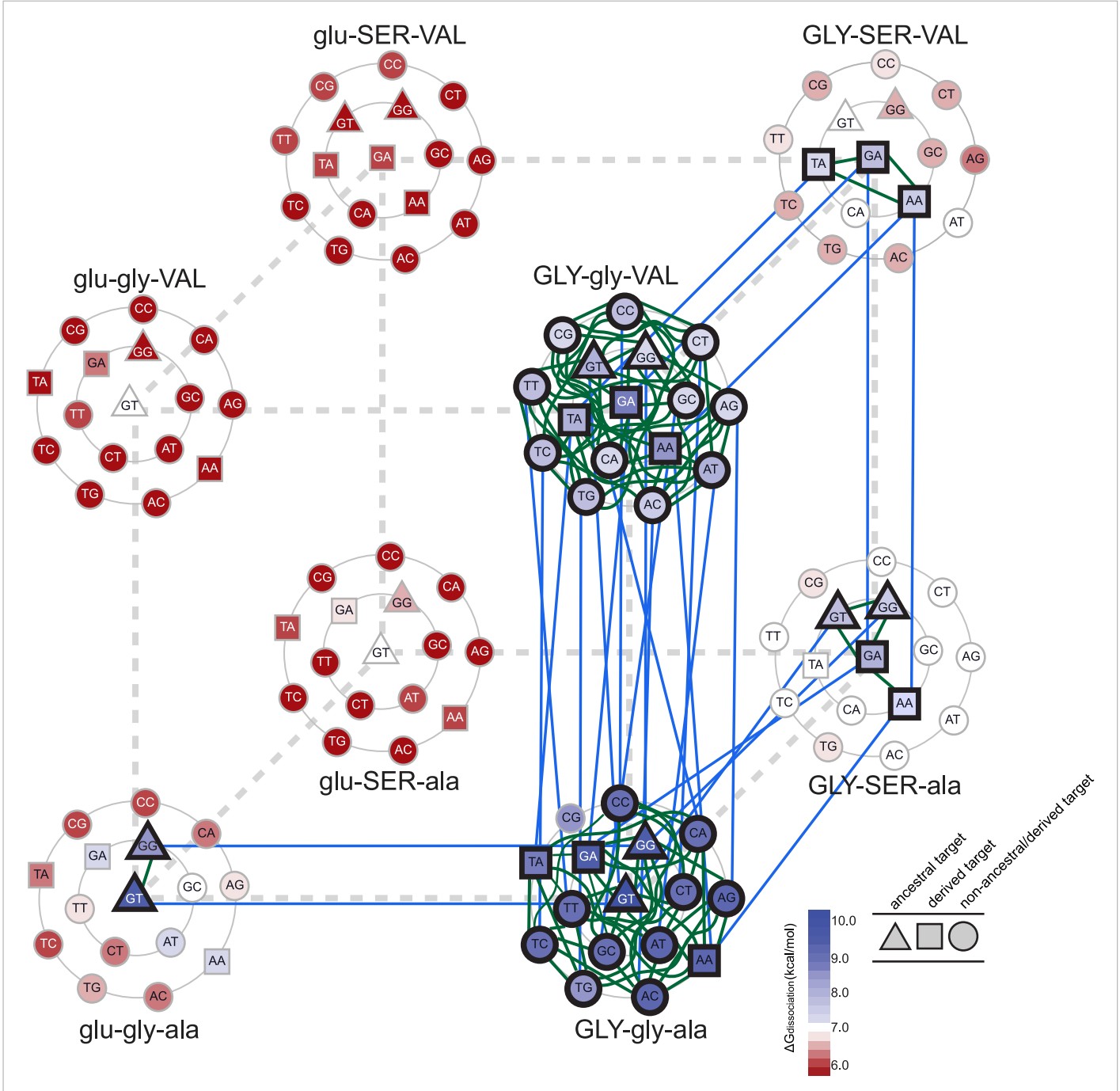

**Figure 5**. Accessible mutational pathways in the joint TF-RE sequence space. Each vertex of the cube represents a protein genotype between AncSR1 and AncSR1+RH; amino acid states at variable RH residues are shown; lower and upper case denote ancestral and derived states, respectively. Each protein's affinity for the 16 REs is shown using the color gradient from red to blue (low to high $\Delta G_{dissociation}$). The genotype at the center of each cluster of REs is that protein's highest-affinity target. REs preferred by AncSR1 or AncSR1-RH are shown in triangles or squares, respectively; circles show other REs. TF-RE complexes with high affinity and occupancy (binding energy greater than the average of all TF-RE complexes and affinity within tenfold of that TF's best target) are outlined in bold. Blue lines represent amino acid replacements in the TF that maintain high-affinity/occupancy binding to a RE; green lines represent nucleotide substitutions in the RE that maintain high-affinity/occupancy binding to a TF. Nodes connected by blue or green lines represent the neutral network between the ancestral TF and its RE targets and the derived TF and its distinct targets.

evolution of the RE, and vice versa. At the starting point of AncSR1:ERE, both the TF and the RE genotypes are highly constrained by the requirement to maintain a high-affinity complex with each other: only a single mutation is available to the DNA (from GT to GG), and only one replacement (glu25GLY) is available to the protein (*Figure 5*). Drift along each of these pathways has very different functional implications: if the G4 mutation in the RE were to occur first, the neutral network available to the protein would remain unchanged, with glu25GLY remaining as the only viable replacement on the trajectory towards AncSR1+RH. But if glu25GLY were to occur first, the set of viable mutations available to the RE would radically expand, with almost any RE genotype being tolerated. Reaplcement glu25GLY in the TF therefore acts as a permissive substitution for drift by the RE, allowing the DNA binding site to explore many new pathways—including mutation to SREs—while still maintaining an association with the TF. Thus, neutral substitutions in the protein can permit previously unavailable moves by the DNA through its neutral network.

Conversely, neutral changes by the RE can permit amino acid replacements in the TF to occur that would otherwise have been unfavorable. For example, if the complex is in the state GLY-SER-ala:GT, adding the final replacement ala29VAL would lead to a nonfunctional complex. But if the RE were to drift first to one of the SREs, then the complex would be able to tolerate this amino acid change. Further, neutral substitutions in the RE can also restrict evolutionary pathways that were previously open to the TF, making certain amino acid changes inaccessible. For example, if the complex is in the state GLY-gly-VAL:GA, the TF can drift to AncSR1+RH via the gly26SER replacement; however, if the RE first drifts to GC, this replacement would abolish the TF:RE association, making it an unlikely evolutionary step (*Figure 5*).

Permissive and restrictive mutations across the interface may act in a serial chain that reciprocally modifies the partner molecules' evolvability: a mutation in the RE may change the protein's capacity to tolerate a previously unavailable replacement, which then changes the RE's tolerance of a mutation that would otherwise have been unavailable, and so on. To follow one such example, replacement glu25GLY allows the ancestral ERE (GT) to drift to TT, which prevents gly26SER, which otherwise would have been available, and also permits the previously denied replacement ala29VAL; if ala29VAL does occur next, then mutation of the DNA to TA becomes available, which in turn is permissive for gly26SER. That replacement closes off many DNA mutations that were previously available, but the final mutation to SRE1 or SRE2 is open. Epistasis across the molecular interface therefore makes the evolution of the TF and the RE contingent upon the genotype—and therefore the prior evolutionary history—of its partner. A chain of serially contingent events may ensue, as permissive and restrictive mutations in the protein and DNA open or close evolutionary paths for mutations within the same molecule or its binding partner.

Taken together, these findings indicate that epistasis across a molecular interface can allow interacting molecules to both evolve by drift to states that are incompatible with the ancestral versions of their partner. AncSR1:ERE is a highly specific complex that does not recognize SRE; it lies on a narrow peninsula in sequence space, with few mutations leading directly to other functional complexes. A series of permissive mutations in both partners, however, could have allowed the RE and the TF to drift together through numerous new genotypes, eventually reaching AncSR1+RH:SRE, which itself is a specific complex on a narrow peninsula that does not bind or connect directly to ERE.

The nonlinear mapping we imposed from biochemical property to biological function/fitness added an additional form of epistasis to that we observed due to affinity alone. The criteria that we used to define this mapping are highly simplified. In reality, the selective effects of mutations that change TF:RE interactions are likely to depend on many factors, including the genomic context and role of other TFs in regulating a given target, interactions between the TF and other proteins, the genotype and function of other domains and sites in the TF, the physiological roles of the target genes, and the demographic characteristics of the population. How these factors affected the mapping of affinity onto fitness for the ancient molecular complex we study here is unknown; as a result, the mutational pathways most likely to have been followed during the evolutionary history of the SR DBD and its targets are also unknown. Despite this uncertainty about specific historical scenarios, however, it is clear that rampant epistasis was present at the most fundamental biochemical level, and even the simple biological criteria we imposed resulted in very strong impacts on the evolutionary accessibility of trajectories across sequence space. It therefore seems likely that epistasis will also structure evolutionary trajectories under more complex, realistic conditions that introduce further nonlinearities into the relationship between physical properties and selectable biological outcomes.

## The biophysical causes of specificity are genotype-specific

Finally, we sought to understand the underlying biophysical mechanisms that cause variation in binding affinity among TF:RE pairs in this region of sequence space. To determine these mechanisms, we performed molecular dynamics (MD) simulations for AncSR1, AncSR1+RH and all intermediate protein genotypes, each bound to every one of the 16 DNA sequences. We then measured hydrogen bonding and packing at the protein-DNA interface, which are known to contribute to high-affinity interactions between proteins and DNA (*von Hippel, 1994*; *Garvie and Wolberger, 2001*; *Coulocheri et al., 2007*; *Rohs et al., 2010*; *McKeown et al., 2014*). For each protein in complex with all 16 REs, we used linear regression to analyze the statistical relationship between each biophysical parameter and affinity.

We found that the number of hydrogen bonds formed across the protein-DNA interface is not positively correlated with the affinity of the TF-RE complex when all 128 combinations are examined (*Figure 6A,C*). Thus, hydrogen bonding does not provide even a partial global explanation of affinity; however, it does explain, in part, affinity for a few specific genotypes. When we separately analyzed each protein's affinity across REs, we found that the number of hydrogen bonds was positively and significantly correlated with affinity for 2 of the 8 protein genotypes (*Figure 6C*, *Figure 6—figure supplement 1*), explaining at most 30% of the variation in affinity. These proteins contain residue glu25, which in the crystal structure of AncSR1:ERE forms hydrogen bonds to specific nucleotide bases in the DNA major groove (*Figure 1C*) (*McKeown et al., 2014*).

Packing also does not provide a global explanation for variation in binding affinity. When all TF:RE combinations were analyzed, the efficiency of packing was not positively correlated with affinity (*Figure 6B,C*). When analyzed separately, packing was significantly and positively associated with affinity for only two proteins, explaining at most 40% of the variance in a protein's affinity across REs (*Figure 6C*, *Figure 6—figure supplement 2*). There is no clear pattern to explain which protein genotypes manifest a correlation of packing with affinity.

We conclude that no simple biophysical metric provides a general explanation of variation in affinity among TF-RE complexes. There may be common mechanisms for the effects of each amino acid replacement on affinity that are common to all REs: for example, replacement gly25GLY improved affinity for every RE, possibly by enhancing the entropic benefit of binding driven by the hydrophobic effect or decreasing the entropic cost of binding by introducing additional degrees of freedom in the protein backbone. But specificity—variation in affinity among REs—appears to be determined by biophysical interactions that are largely unique to each TF-RE combination. For example, replacement ala29VAL improves relative affinity for REs containing A4; in previously published ancestral X-ray crystal structures, the hydrophobic side chain of VAL29 packs against the hydrophobic methyl groups that are unique to the complementary T at position 4 in the RE, providing a likely explanation, at least in part, for this effect (*McKeown et al., 2014*). These interactions only affect packing involving a small number of atoms, and they are present in only a small number of TF-RE pairs, so they do not have a statistically detectable effect on the relationship between packing and affinity across all REs.

These results reinforce the importance of epistasis across the molecular interface. Differences in binding affinity among TF:RE complexes are typically determined by unique physical interactions between atoms on the protein and atoms on the DNA, and higher-order interactions sometimes involve more than two atoms across the interface.

## Discussion

### Methodologies for characterizing epistasis in combined sequence space

Linear modeling strategies have previously been used to statistically characterize the main and epistatic effects of variation in a DNA sequence on affinity for a specific TF, typically using high-throughput approaches that estimate affinity from measurements of occupancy (*Benos et al., 2002*; *Stormo, 2011*; *Zhao et al., 2012*; *Stormo et al., 2015*). We extended these methods, using direct measurements of affinity for every combination of TF and RE genotypes in a defined region of sequence space, to identify the genetic determinants of protein-DNA affinity both within and between the two molecules.

This strategy has both advantages and limitations. By simultaneously studying the effects of variation within the RE, within the TF, and between the two molecules were we able to describe in detail how changes in the TF protein affect the specificity of RE binding and vice versa. This approach

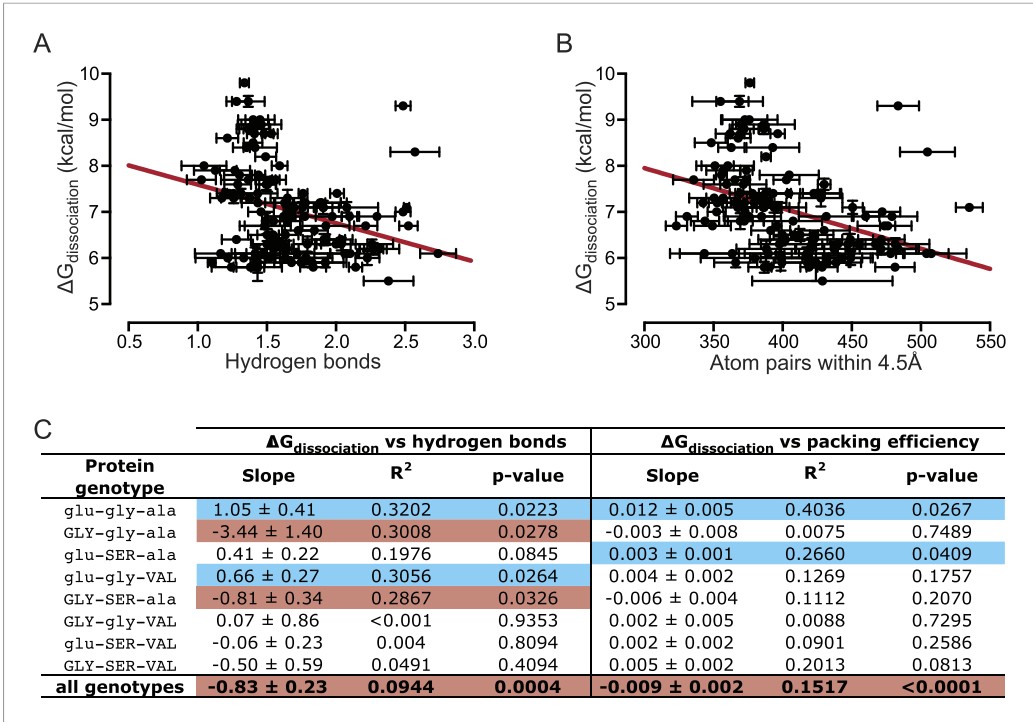

Figure 6. Hydrogen bonding and packing efficiency do not explain TF-RE affinity. (**A**) The number of hydrogen bonds formed between atoms in the RH and atoms in the RE in molecular dynamic (MD) simulations is not positively correlated with the experimentally measured binding energy of TF-RE complexes. Each data point represents the number of hydrogen bonds formed by one of the 128 TF-RE pairs (8 variants of AncSR1 with 16 variant REs), each averaged over three replicate 50 ns simulations; error bars show SEM. Red line indicates best-fit linear regression model. For p-value and $R^2$, see panel **C**. (**B**) The efficiency of packing interactions across the RH-RE interface in MD simulations is not positively correlated with the experimentally measured binding energy of TF-RE complexes. In MD simulations, the number of protein-DNA atom pairs within 4.5 Å of each other was calculated for all 128 TF:RE complexes. Points and error bars show the mean and SEM over three replicate MD simulations. Red line indicates best-fit linear regression model; p-value and $R^2$ are shown in **C**. (**C**) Correlation of hydrogen bonding and packing efficiency with binding energy for individual protein genotypes. For each TF, the experimentally measured binding energy for each of the 8 REs was regressed against either the number of hydrogen bonds formed from RH to RE or the efficiency of packing between RH and RE. The presence of positive (blue), negative (red), or non-significant (NS) correlations is indicated, along with the p-value of the correlation and the fraction of variation in binding energy explained by each dependent variable ($R^2$). For full data sets and regressions, see *Supplementary file 1*.

The following figure supplements are available for figure 6:

**Figure supplement 1**. Direct hydrogen bonding at the protein-DNA interface positively correlates with binding affinity for only 2 out of 8 protein genotypes.

**Figure supplement 2**. Packing efficiency at the protein-DNA interface positively correlates with binding affinity for only 2 out of 8 protein genotypes.

can in principle be used to study interactions in joint sequence space for any complex of molecules, including those with more than two elements. It can also be applied to large datasets, larger numbers of sites, and higher-order interactions, so long as the data are sufficiently precise to allow robust quantitative analysis.

Our experimental approach allowed us to obtain precise and direct measurements of affinity for each individual TF-RE complex, which were essential to our analysis. Statistical models of higher-order epistasis across a protein-DNA interface add many terms to the regression models previously used for individual proteins (*Zhao et al., 2012*; *Stormo et al., 2015*). As models become complex, estimates of

higher-order determinants of affinity can become uncertain or biased if the estimates of affinity are noisy or imprecise; this problem can be exacerbated if affinities for some sequence states are estimated by interpolation rather than being measured (*Otwinowski and Plotkin, 2014*). By directly and precisely measuring the ΔG of binding for all TF:RE combinations, we were able to detect and estimate the magnitude and importance of higher-order epistatic effects with relatively high confidence. Further, because effects on the free energy of binding by independent factors are additive for reversible interactions, these measurements allowed us to use linear regression to rigorously estimate the magnitude and significance of main and epistatic effects on TF-RE recognition. Other metrics such as occupancy or entropy may require more complex descriptive models.

Obtaining such precise and direct measurements of binding energy is time consuming and limits the number of complexes that can be studied. The 128 TF:RE combined genotypes whose affinity we measured represent a tiny slice of the vast joint sequence space of possible TFs and REs. Even within this small region of sequence space, however, we found rampant epistatic interactions within the DNA RE, within the TF protein, and between the RE and the TF. These epistatic interactions play key roles in determining the specificity of binding by each TF, and they strongly shape the functional topology of the joint sequence space and the capacity of each molecule to traverse this space under various evolutionary scenarios. As improving technical capacities allow larger tracts of joint sequence space to be analyzed and higher-level biological functions to be assessed, it seems quite likely that epistasis will be a major influence on the specificity and evolution of molecular complexes.

## Epistasis and the evolution of molecular complexes

Complexity and interdependence are often thought to act as constraints on evolution (*Lewontin, 1984*; *Bonner, 1988*; *Kauffman, 1993*; *Wagner and Altenberg, 1996*; *Orr, 2000*; *Schank and Wimsatt, 2001*). Previous studies of epistasis within a molecule have shown that interactions among mutations constrict the number of passable evolutionary trajectories through sequence space and make the outcomes of evolution contingent upon the prior occurrence of permissive mutations (*Weinreich et al., 2006*; *Bridgham et al., 2009*; *Podgornaia and Laub, 2015*). The joint sequence space of multiple interacting molecules contains more dimensions and therefore far more paths between functional joint genotypes than does either molecule's separate sequence space (*Gavrilets, 2004*). Our work shows that intermolecular epistasis is indeed rampant and blocks many of these additional pathways.

But intermolecular epistasis also has an opposite effect, which introduces new degrees of freedom into the evolutionary process. Permissive epistatic mutations across the molecular interface open paths for the other molecule that would otherwise have been blocked. As a result, the absolute number of trajectories (and ultimate endpoints) available to either partner is larger than it would appear to be if only the 'slice' or profile through the joint sequence space represented by variability in a single molecule were considered. Under the scenario we examined, for example, an ERE regulated by an immutable AncSR1-DBD could not reach SRE without losing functionality, and AncSR1 could not reach AncSR1+RH if it regulated only immutable binding sites. But because amino acid replacements in the TF are permissive for mutations in the RE, and subsequent RE mutations are permissive for replacements in the protein, both members of the complex can explore regions of their own sequence space and reach genotypes that were not previously available to either.

Molecular complexes are pervasive in biology. The extensive intermolecular epistasis that we observed among a small number of sites in a simple binary complex suggests that the reach of intermolecular epistasis may be vast. If the structure of that epistasis is anything like that present in the AncSR1-RE complex, then the web of contingent events that structures evolutionary trajectories is likely to be dense, with the evolutionary potential of any one molecule depending on prior events in other molecules. But if some of those events are permissive, as our results suggest they are likely to be, then the potential of evolution to generate new functional complexes by the combined action of mutation, drift and selection is greater than it may appear if the molecules are viewed only in isolation. Complexity and interdependence not only constrain evolution; they can also buy freedom for the parts of a system to reach new states, if the historical events are right.

# Materials and methods

## Protein purification

DBDs were cloned into the pETMALc-H10T vector (*Pryor and Leiting, 1997*) (a gift from John Sondek, UNC-Chapel Hill) C-terminal to a cassette containing a 6xHis tag, maltose binding protein (MBP) and a TEV protease cleavage site. DBDs were expressed in BL21(DE3)pLysS Rosetta cells. Protein expression was induced by addition of 1 mM IPTG at $A_{600}$ of 0.8–1.2. After induction, cells were grown overnight at 15°C. Cells were harvested via centrifugation and frozen at −10°C overnight. Cells were lysed using B-PER Protein Extraction Reagent Kit (Thermo Scientific).

Lysate was loaded onto a pre-equilibrated 5 ml HisTrap HP column (GE Fairfield, CT) and eluted with a linear imidazole gradient (25 mM–1 M) in 25 mM sodium phosphate and 100 mM NaCl buffer [pH 7.6]. The DBD was cleaved from the MBP-His fusion using TEV protease in dialysis buffer consisting of 25 mM sodium phosphate, 150 mM NaCl, 2 mM βME and 10% glycerol [pH 8.0]. The cleavage products were loaded onto a 5 ml HiPrep SP FF cation exchange column (GE) and eluted with a linear NaCl gradient (150 mM–1 M) in 25 mM sodium phosphate buffer [pH 8.0]. DBDs were further purified on a Superdex 200 10/300 GL size exclusion column (GE) with 10 mM Tris [pH 7.6], 100 mM NaCl, 2 mM βME, 5% glycerol. Protein purity was assayed after each purification by visualization on a 12% SDS-PAGE gel stained with Bio-Safe Coomassie G-250 stain (Bio-Rad).

## FA binding assay

DNA constructs were ordered from Eurofins Operon (Huntsville, AL) as HPLC-purified single stranded oligos with the forward strand labeled at the 5′-end with 6-FAM. Sequences of forward strands, with differences underlined, were as follows: CCAG<u>GC</u>CA, CCAG<u>GG</u>CA, CCAG<u>CT</u>CA, CCAG<u>CA</u>CA, CCAG<u>CC</u>CA, CCAG<u>CG</u>CA, CCAG<u>TT</u>CA, CCAG<u>TA</u>CA, CCAG<u>TC</u>CA, CCAG<u>TG</u>CA, CCAG<u>AC</u>CA, CCAG<u>AG</u>CA, CCAG<u>GT</u>CA, CCAG<u>AA</u>CA, CCAG<u>GA</u>CA, CCAG<u>AT</u>CA. Complementary reverse strands were also ordered.

Forward and reverse strands were re-suspended in duplex buffer (30 mM Hepes [pH 8.0], 100 mM potassium acetate) to a concentration of 100 μM. Equimolar quantities of complementary forward and reverse strands were combined and placed in a 95°C water bath for 10 min then slowly cooled to room temperature. The double stranded product was diluted to 5 μM in water.

Purified DBD was buffer exchanged using Illustra NAP-25 columns into 20 mM Tris [pH 7.6], 130 mM NaCl and 5% glycerol. Protein concentration was determined by measuring absorbance at 280 nm, 320 nm and 340 nm and correcting for light scattering. A range of DBD concentrations was titrated in triplicate onto a black, NBS-coated 384 well plate (Corning 3575, Corning, NY). Labeled DNA was added to each well to achieve a final concentration of 5 nM in 91 μl total volume. Sample FP was read using a Perkin Elmer Victor X5, exciting at 495 nm and measuring emission anisotropy at 520 nm. To determine $K_1$, we measured binding affinity to the half-site REs in triplicate and fit the data to a single-site binding model.

## Linear modeling the genetic determinants of binding affinity

### Definition of genetic encoding system

To quantify the genetic determinants of binding affinity, we used an approach similar to that previously developed (*Stormo, 2011*). We constructed regression models that explain $\Delta G_{dissociation}$ as a function of the genetic states at the three variable amino acid residues in the protein and the two variable nucleotide positions in the RE half-site.

The genetic variation in the protein was defined in the linear models using one-dimensional variables for the RH substitutions; residues 25, 26 and 29 were are described by single-dimensional vectors a, b, and c, respectively, with the ancestral state defined as −1 and the derived state defined as +1 (*Supplementary file 2*). For the two variable nucleotide positions in the RE, each of the four alternate genetic states was represented using the *WYK* tetrahedral-encoding system (*Zhang and Zhang, 1991*) in which A, C, G, and T are represented as different linear vectors in a three-dimensional space defined by $[w_i \, y_i \, k_i]$ (where $i$ indicates the site in the RE). A is represented by the linear vector (1, −1, −1), C by (−1, 1, −1), G by (−1, −1, 1), and T by (1, 1, 1) (*Supplementary file 2*). These variables make the y-intercept of the linear model equal to the mean $\Delta G_{dissociation}$ across all experimental measurements (*Stormo, 2011*); therefore, all genetic effects are expressed relative to the mean.

## First-order linear models

We constructed our first-order model by regressing the $\Delta G_{dissociation}$ of each genotype on dependent variables that reflect the individual first-order identities at each genetic position. For example, the linear model for position 25 in the RH is expressed as

$$(\Delta G_{dissociation}) = a(u_1) + u_0,$$

where $a$ is the effect coefficient of moving $+1$ in that dimension, $u_1$ is the coordinate representing the genotype (i.e., $-1$ for ancestral glu, $+1$ for derived GLY), and $u_0$ is the y-intercept for the model (equal to the mean across the data). The linear coefficients for each model were computed using ordinary least squares regression with the open-source statistical package R (http://www.r-project.org/). The coefficient $a$ indicates the deviation of the derived genetic state from the mean, while $-a$ gives the deviation of the ancestral genetic state from the mean. The total effect of the substitution glu25GLY is therefore equal to $2a$.

To determine how well all three first-order effects of substitutions in the protein predict variation in $\Delta G_{dissociation}$, we constructed the following linear model that included all first-order protein coefficients:

$$(\Delta G_{dissociation}) = a(u_1) + b(u_2) + c(u_3) + u_0,$$

where $u_2$ and $u_3$ are the coordinates representing the genotype for position 26 and 29, respectively. We then computed the $R^2$ and adjusted $R^2$ values for this protein-only first-order model (*Figure 3*, *Supplementary file 1*).

The site-specific first-order models for each site in the RE were modeled in terms of the w, y, and k coefficients. For example, the linear model relating $\Delta G_{dissociation}$ to variation in the third position is expressed as

$$(\Delta G_{dissociation}) = w_3(u_4) + y_3(u_5) + k_3(u_6) + u_0,$$

where $u_4$, $u_5$, and $u_6$ are the coordinates representing the genotype at site 3 in the RE, while $w_3$, $y_3$, and $k_3$ are the coefficients for the effect on $\Delta G_{dissociation}$ per unit in $w_3$, $y_3$, and $k_3$ space. The magnitude of the effect of each genotype on $\Delta G_{dissociation}$ was determined by computing the sum of the modeled *WYK* coefficients for its defined coordinates in $w_3$, $y_3$, and $k_3$ space (e.g., the effect of C in position 3 is equal to $-w_3 + y_3 - k_3$: See *Supplementary file 2*). To determine how well the first-order effects of RE variables predict variation in $\Delta G_{dissociation}$, we constructed the following linear model:

$$(\Delta G_{dissociation}) = w_3(u_4) + y_3(u_5) + k_3(u_6) + w_4(u_7) + y_4(u_8) + k_4(u_9) + u_0.$$

We then computed the $R^2$ and adjusted $R^2$ values for this RE-only first-order model (*Supplementary file 1*). In modeling the genetic effects within the RE for each protein genotype, we also performed a likelihood ratio test in order to assess whether adding each site's first-order effects significantly improved the fit to a model only including first-order variation at the other site (e.g., to assess the importance of site 3, we compared the fit of a model containing terms for site 4 vs a model containing terms for both site 3 and site 4; if the more complex model was significant by likelihood ratio test, this indicates that variation at site 3 is significantly predictive of binding affinity).

We also constructed a global first-order linear model that included both RH and RE variables. The global first-order model is as follows:

$$(\Delta G_{dissociation}) = a(u_1) + b(u_2) + c(u_3) + w_3(u_4) + y_3(u_5) + k_3(u_6) + w_4(u_7) + y_4(u_8) + k_4(u_9) + u_0.$$

To determine how well the combined first-order effects of both the protein and RE variables predicted variation in $\Delta G_{dissociation}$, we determined the $R^2$ and adjusted $R^2$ values (*Supplementary file 1*), as well as performing likelihood ratio tests for each additional set of parameters. Together, we refer to these as *ABC-/WYK- encoded linear models*.

## Linear models with second-order intra-molecular epistasis

To identify cases of second-order epistatic interactions, we individually introduced every possible interaction term for every two-way combination of genotypes at the variable sites in the protein or the RE. These interaction variables were constructed as previously described (*Stormo, 2011*). Each

interaction is described by a new linear vector, the value for which is determined by taking the outer product between the two first-order linear vectors. For example, the interaction between site 25 and 29 of the RH will be equal to $(a) \otimes (c) = (ac)$. The linear model that includes both first and second-order terms in the protein is as follows:

$$(\Delta G_{dissociation}) = a(u_1) + b(u_2) + c(u_3) + ab(u_{10}) + ac(u_{11}) + bc(u_{12}) + u_0,$$

where $u_{10}$ is equal to $u_1 u_2$, and so on. The second-order interaction effects are equal to the deviation from the additive effect modeled by each genetic state individually across other genetic backgrounds, and is defined herein as the 'marginal' effect. As previously described, this method of encoding means that all terms are relative to the mean $\Delta G_{dissociation}$, and so we multiplied this term by two in order to obtain the effect of each substitution in the derived genetic background (i.e., when the derived state exists at the other site) vs the average effect of that substitution regardless of the genetic background (e.g., interaction 25_29 is equal to $2ac$). The calculation of these terms is summarized in *Supplementary file 2*.

Interactions between each site in the RE were modeled analogously. Here, the interaction between site 3 and site 4 in the RE is constructed by: $(w_3, y_3, k_3) \otimes (w_4, y_4, k_4) = (w_3 w_4, w_3 y_4, w_3 k_4, y_3 w_4, y_3 y_4, y_3 k_4, k_3 w_4, k_3 y_4, k_3 k_4)$. The linear model that includes both first and second-order terms in the RE is as follows:

$$\begin{aligned}(\Delta G_{dissociation}) = \; & w_3(u_4) + y_3(u_5) + k_3(u_6) + w_4(u_7) + y_4(u_8) + k_4(u_9) + w_3 w_4(u_{13}) \\ & + w_3 y_4(u_{14}) + w_3 k_4(u_{15}) + y_3 w_4(u_{16}) + y_3 y_4(u_{17}) + y_3 k_4(u_{18}) \\ & + k_3 w_4(u_{19}) + k_3 y_4(u_{20}) + k_3 k_4(u_{21}) + u_0.\end{aligned}$$

One advantage of this method of encoding the genetic data is that the first-order model is nested within the second-order model. This allowed us to assess whether addition of the second-order model terms significantly improved the fit by comparing the improvement in the adjusted $R^2$ as well as the improvement in the likelihood ratio test relative to the simpler first-order model. The effect of each second-order interaction (i.e., the epistasis that should be added to the sum of the additive lower-order effects) can be solved from these coefficients. For example, the epistatic interaction between C at position 3 and A at position 4 is:

$$C3\_A4 = -w_3 w_4 + w_3 y_4 + w_3 k_4 + y_3 w_4 - y_3 y_4 - y_3 k_4 - k_3 w_4 + k_3 y_4 + k_3 k_4.$$

To identify the genetic determinants of binding affinity for both the protein and RE, we applied this analysis to each individual protein genotype (*Supplementary file 1*), as well as globally across all protein and RE genotypes (*Supplementary file 2*).

## Linear models with second-order inter-molecular epistasis

We next constructed a linear model that included both first- and second-order terms within the protein and RE as well as second-order terms between the protein and RE (i.e., interactions between specific amino acid and nucleotide states). These second-order interaction terms between the protein and RE were determined in the same manner as before. For example, the interaction between the substitution at site 25 in the protein and C at site 3 in the RE is determined by adding the terms given by: $(a) \otimes (-w_3, y_3, -k_3) = (-aw_3, ay_3, -ak_3)$ (*Supplementary file 2*). We added these inter-molecular second order terms to the previously described model that included all first- and second-order intra-molecular terms. Significance of the more complex model was assessed by its improvement of adjusted R-squared and by likelihood ratio test. For all terms resulting from this model, see *Supplementary file 2*.

## Third-order epistatic linear models

We next constructed a linear model that included third-order interactions. These third-order terms described the interactions between one state in the protein and a pair of states in the RE (and vice versa). These interactions were modeled analogously to the second-order terms, by determining the outer product between all relevant lower-order terms.

The significance of the improvement in the fit resulting from the addition of these third-order interaction terms was assessed by how much the set of third-order terms improved the adjusted $R^2$ as well as by likelihood ratio test (*Supplementary file 1*). The script for these analyses is available from Github: github.com/danderso8/eLife2015_LinearModelingandMolecularDynamics.

## Validating with an alternative genetic encoding system

Finally, we wanted to confirm that the effects we modeled using these methods were consistent with other modeling approaches. We constructed analogous first-, second-, and third-order models using a binary encoding system, in which each variable is 1 if the respective genetic state is at a given position, and 0 otherwise (e.g., glu25 is 1 if there is a glu at position 25, and 0 in all other cases). For higher-order models, the terms were constructed so that they equal 1 if all lower-order genetic states are present (e.g., glu25_G3 is 1 if both site 25 in the RH is a glu and site 3 in the RE is a G, and 0 otherwise). We constructed these linear models by determining the effect of each binary variable relative to the mean $\Delta G_{dissociation}$. We then compared the effect of all the same first- and second-order genetic effects with this approach to our results from *abc-/WYK*-models and found they were entirely consistent, as expected (*Stormo, 2011*; *Poelwijk et al., 2015*).

## MD simulations

The crystal structure of AncSR1 bound to ERE (PDB: 4OLN) was used as the starting point for all simulations. Historical substitutions and changes to the DNA RE sequences were introduced in silico (*Emsley and Cowtan, 2004*). Each system was solvated in a cubic box with a 10 Å margin, then neutralized and brought to 150 mM ionic strength with sodium and chloride ions. This was followed by energy minimization to remove clashes, assignment of initial velocities from a Maxwell distribution, and 1 ns of solvent equilibration in which the positions of heavy protein and DNA atoms were restrained. Production runs were 50 ns, with the initial 10 ns excluded as burn-in. The trajectory time step was 2 fs, and final analyses were performed on frames taken every 12.5 ps.

We used TIP3P waters and the AMBER FF03 parameters for protein and DNA, as implemented in GROMACS 4.5.5 (*Duan et al., 2003*). The zinc fingers were treated with a recently derived bonded potential for Cys-Zn interactions (*Hoops and Rindler, 1991*; *Lin and Wang, 2010*) as previously described (*McKeown et al., 2014*). Zinc finger partial charges were derived using the RED III.4 pipeline (*Dupradeau et al., 2010*) as previously described (*McKeown et al., 2014*). We extracted a tetrahedral Cys$_4$ zinc finger from a 0.9 Å crystal structure (*Iwase et al., 2011*), optimized its geometry with an explicit quantum mechanical calculation using the 6–31G** basis set (*Schuchardt et al., 2007*), then derived partial charges using RESP (*Dupradeau et al., 2010*). All quantum mechanical calculations were performed using the FIREFLY implementation of GAMESS (*Schmidt and Mohring, 1993*; *Granovsky, 2007*,). We verified that the zinc fingers maintained their tetrahedral geometry over the course of the simulations.

Simulations were performed in the NTP ensemble at 300 K, 1 bar. All bonds were treated as constraints and fixed using LINCS (*Hess et al., 1997*). Electrostatics were treated with the Particle Mesh Ewald model (*Darden and Pedersen, 1993*), using an FFT spacing of 12 Å, interpolation order of 4, tolerance of 1e-5, and a Coulomb cutoff of 9 Å. van der Waals forces were treated with a simple cutoff at 9 Å. We used velocity rescaled temperature coupling with a $\tau$ of 0.1 ps and Berendsen pressure coupling with a $\tau$ of 0.5 ps and a compressibility of 4.5e-5 bar$^{-1}$. Analyses were performed using VMD 1.9.1 (*Humphrey et al., 1996*)—with its built-in TCL scripting utility—as well as a set of in-house Python scripts (adapted from scripts generously shared by Mike Harms, and available from Github: github.com/harmsm/md-analysis-tools).

## Acknowledgements

We thank Patrick Phillips, Mike Harms, Tyler Starr, Jamie Bridgham, and members of the Thornton Lab for advice and comments. The University of Oregon ACISS cluster provided computing resources. This work was supported by NIH R01-GM104397 (JWT), NIH training grant 5-T32-GM-7759-33 (ANM), AHA award 11PRE7510085 (DWA), and a HHMI Early Career Scientist Award (JWT).

## Additional information

### Funding

| Funder | Grant reference | Author |
| --- | --- | --- |
| American Heart Association (AHA) | 11PRE7510085 | Dave W Anderson |

| Funder | Grant reference | Author |
|--------|-----------------|--------|
| National Institutes of Health (NIH) | R01-GM104397 | Joseph W Thornton |
| National Institutes of Health (NIH) | 5-T32-GM-7759-33 | Alesia N McKeown |
| Howard Hughes Medical Institute | Early Career Scientist Award | Joseph W Thornton |

The funders had no role in study design, data collection and interpretation, or the decision to submit the work for publication.

### Author contributions

DWA, ANM, Conception and design, Acquisition of data, Analysis and interpretation of data, Drafting or revising the article; JWT, Conception and design, Analysis and interpretation of data, Drafting or revising the article

# Additional files

### Supplementary files

• Supplementary file 1. First-order and epistatic genetic determinants of binding affinity. First-order effects indicate the difference in binding energy relative to the mean across all data, while the second-order effects are the marginal addition to the additive sum of the first-order effects. Third-order effects are the marginal addition to the additive sum of all lower-order effects. (**A**) The energetic effects of binding for all first-order and epistatic terms in the RE as determined by linear modeling for each protein genotype. (**B**) The energetic effects for amino acid replacements averaged across all 16 REs. (**C**) The energetic effects from a global model, including all possible first-, second-, and third-order effects within and between the protein and DNA.

• Supplementary file 2. abc/WYK- encoding of sequence characters for linear modeling of genetic effects. (**A**) One-dimensional vectors for ancestral versus derived state at variable amino acid positions 25, 26, and 29 in the protein are shown. (**B**) Three-dimensional vectors for A, C, G, or T at variable positions 3 and 4 in the RE are shown. The encoding methods shown in panels **A** and **B** ensure that the origin in each vector space will be associated with the mean value of the independent variable (in this case, the delta-G of dissociation) across all the data. (**C**) Terms used in the linear model using abc/WYK coding. Each row shows the expression for the effect on the independent variable of a nucleotide state, amino acid replacement, or interaction among them. Each genetic effect is calculated using the expression shown and the optimized values of the linear coefficients as described in 'Materials and methods'.

### Major dataset

The following previously published dataset was used:

| Author(s) | Year | Dataset title | Dataset ID and/or URL | Database, license, and accessibility information |
|-----------|------|---------------|-----------------------|--------------------------------------------------|
| McKeown AN, Bridgham JT, Anderson DW, Murphy MN, Ortlund EA, Thornton JW | 2014 | Ancestral Steroid Receptor 1 in complex with estrogen response element DNA | http://www.rcsb.org/pdb/explore/explore.do?structureId=4OLN | Publicly available at RCSB Protein Data Bank (Accession No: 4OLN). |

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
