## [Decision Letter]

Thank you for submitting your work entitled “Epistasis Shaped the Evolutionary Sequence Space of an Ancestral Transcription Factor and its DNA Regulatory Elements” for peer review at *eLife*. Your submission has been favorably evaluated by Detlef Weigel (Senior editor) and three outside reviewers.

The reviewers have discussed the reviews with one another and the editor has drafted this decision to help you prepare a revised submission. There was broad agreement that this is a creative and important study, but the reviewers also made several suggestions how the impact of the work could be improved.

Specific recommendations include:

1) Acknowledge that the model (single monomer and focusing on just two positions in the DNA sequence) is probably an oversimplification, and brief discussion of potential caveats that arise from this.

2) Provide a better justification of the use of a linear model. Currently, there is a single sentence in the manuscript; you seem to be using the linear model to expose epistasis as deviations from the linear model's assumption of additivity, but this is poorly explained in the text. The limitations of the linear model are discussed in references (20; 86; 9) but these could easily be summarized in the text.

3) Justify why you consider only two positions in the DNA sequence; some references are given in the current test, but the discussion is brief to the point of confusion.

4) Provide a clearer justification of your threshold for functionality. While is seems unlikely that the proteins would evolve through non-functional intermediates (even after gene duplication), your arguments would be strengthened by a more thorough discussion if you chose the particular thresholds.

5) Cite theoretical background for dependence between nucleotide positions (Stormo lab work and others, HMM based models) and papers describing that offer examples of interdependence between nucleotides or amino acid positions. The editor's favorite protein, LEAFY, is one among many others.

6) Describe results and modeling in a more intuitive manner that can be more easily followed. Clearly distinguish entirely new concepts from old ideas being addressed in a fresh manner.

7) Provide a more accurate summary in the introduction of the non-additivity of interactions at a protein-DNA interface ([67], JMB 301, 597-624).

---

## [Author Response]

*1) Acknowledge that the model (single monomer and focusing on just two positions in the DNA sequence) is probably an oversimplification, and brief discussion of potential caveats that arise from this*.

*3) Justify why you consider only two positions in the DNA sequence; some references are given in the current test, but the discussion is brief to the point of confusion*.

1 and 3. We have acknowledged that the system we used—focusing on two positions in the DNA sequence of a half-site—represents a very limited portion of the DNA sequence space that the evolving TF-RE system could in principle explore, and we have discussed the implications of this choice. This part of the space represents the historically relevant portion of the sequence space, because these are the only two positions that vary between the ancestrally preferred RE and the derived preferred RE, and only half-site specificity changed across the functional transition from AncSR1 to AncSR1+RH—not cooperativity or other aspects of dimeric binding. Limiting our study to a relatively small region of multidimensional joint protein+DNA sequence space was necessary for us to exhaustively map function and quantify genetic factors contributing to variation in affinity (we explore these points in the following subsections of the text: “Epistasis across a molecular interface: transcription factors and DNA response elements”, “Exploitation of latent binding affinity” and “Methodologies for characterizing epistasis in combined sequence space”).

*2) Provide a better justification of the use of a linear model. Currently, there is a single sentence in the manuscript; you seem to be using the linear model to expose epistasis as deviations from the linear model's assumption of additivity, but this is poorly explained in the text. The limitations of the linear model are discussed in references (20; 86; 9) but these could easily be summarized in the text*.

We clarified the nature and justification for the linear regression model. The model does not assume that only additive effects of individual sites in the protein and DNA determine affinity. Rather, it models the free energy of binding as the sum of the energetic contributions of each individual site’s main effect plus those due to every epistatic interaction. Using a linear model is appropriate based on first principles given the fact that total free energy of a system by definition is the sum of the free energy of the factors that contribute to it. The advantage of this model is that it allows the energetic contributions of epistatic interactions to binding to be precisely quantified and the statistical significance of such contributions to be evaluated (please see the subsections “Determinants of affinity in RE sequence space”, “Effects of amino acid substitutions on affinity” and “Epistasis across a molecular interface”).

4) Provide a clearer justification of your threshold for functionality. While is seems unlikely that the proteins would evolve through non-functional intermediates (even after gene duplication), your arguments would be strengthened by a more thorough discussion if you chose the particular thresholds.

We clarified the biological justification for and limitations of the criteria we used to define functional TF-RE complexes in the latter part of the analysis. Although the criteria are simple, they are biologically motivated, and the fact that we observed extensive epistasis under such a simple nonlinear mapping suggests that more complex forms of nonlinearity, as are likely to occur in real biological systems, are also likely to give rise to epistasis. We also explicitly discussed why gene duplication almost certainly did not release the evolving AncSR1 protein from purifying selection (please refer to the subsections entitled “Mutational pathways to new transcriptional modules” and “Intermolecular epistasis allows coevolutionary drift by the TF-RE complex”).

*5) Cite theoretical background for dependence between nucleotide positions (Stormo lab work and others, HMM based models) and papers describing that offer examples of interdependence between nucleotides or amino acid positions. The editor's favorite protein, LEAFY, is one among many others*.

*7) Provide a more accurate summary in the introduction of the non-additivity of interactions at a protein-DNA interface (*[67]*, JMB 301, 597-624))*.

5 and 7. We modified the text to note that prior research has identified epistasis between sites in TF-RE binding, and that this is expected, given the complex and integrated nature of the protein-DNA interface, citing [67] on this point. We also clarified that the model is an extension of previous work used to analyze the specificity of a TF’s interactions with DNA (please see the subsections headed “Epistasis across a molecular interface: transcription factors and DNA response elements” and “Methodologies for characterizing epistasis in combined sequence space”).

*6) Describe results and modeling in a more intuitive manner that can be more easily followed. Clearly distinguish entirely new concepts from old ideas being addressed in a fresh manner*.

We edited the text to make the Methods and Results easier to follow, including the description of the modeling approach. We also sought to explain more clearly the concepts of sequence space and its functional topology (please see “Function and evolution in molecular sequence space”, “Determinants of affinity in RE sequence space” “Effects of amino acid substitutions on affinity” and “Epistasis across a molecular interface”).